# Transport Stress Induces Oxidative Stress and Immune Response in Juvenile Largemouth Bass (*Micropterus salmoides*): Analysis of Oxidative and Immunological Parameters and the Gut Microbiome

**DOI:** 10.3390/antiox12010157

**Published:** 2023-01-09

**Authors:** Qingchun Wang, Wei Ye, Yifan Tao, Yan Li, Siqi Lu, Pao Xu, Jun Qiang

**Affiliations:** 1Wuxi Fisheries College, Nanjing Agricultural University, Wuxi 214081, China; 2Key Laboratory of Freshwater Fisheries and Germplasm Resources Utilization, Ministry of Agriculture and Rural Affairs, Freshwater Fisheries Research Center, Chinese Academy of Fishery Sciences, Wuxi 214081, China

**Keywords:** largemouth bass, transport stress, gut microbiome, oxidative stress, innate immunity response

## Abstract

Transport is essential in cross-regional culturing of juvenile fish. Largemouth bass (*Micropterus salmoides*) often exhibit decreased vitality and are susceptible to disease after transportation. To study the effects of transport stress on juvenile largemouth bass, juveniles (average length: 8.42 ± 0.44 cm, average weight 10.26 ± 0.32 g) were subjected to a 12 h simulated transport, then subsequently, allowed to recover for 5 d. Liver and intestinal tissues were collected at 0, 6 and 12 h after transport stress and after 5 d of recovery. Oxidative and immunological parameters and the gut microbiome were analyzed. Hepatocytic vacuolization and shortened intestinal villi in the bass indicated liver and intestinal damage due to transport stress. Superoxide dismutase, lysozyme and complement C3 activities were significantly increased during transport stress (*p* < 0.05), indicating that transport stress resulted in oxidative stress and altered innate immune responses in the bass. With the transport stress, the malondialdehyde content first increased, then significantly decreased (*p* < 0.05) and showed an increasing trend in the recovery group. 16S rDNA analysis revealed that transport stress strongly affected the gut microbial compositions, mainly among Proteobacteria, Firmicutes, Cyanobacteria and Spirochaetes. The Proteobacteria abundance increased significantly after transport. The Kyoto Encyclopedia of Genes and Genomes functional analysis revealed that most gut microbes played roles in membrane transport, cell replication and repair. Correlation analyses demonstrated that the dominant genera varied significantly and participated in the measured physiological parameter changes. With 5 days of recovery after 12 h of transport stress, the physiological parameters and gut microbiome differed significantly between the experimental and control groups. These results provide a reference and basis for studying transport-stress-induced oxidative and immune mechanisms in juvenile largemouth bass to help optimize juvenile largemouth bass transportation.

## 1. Introduction

Transportation is important in cross-regional fingerling cultures. However, the fish often showed decreased vitality, loss of appetite, and low resistance after long-distance transportation, which adversely affects healthy breeding and limits rapid development in the aquaculture industry. Many reasons, such as water quality changes, fish body bruising, starvation and so on, may cause transport stress. Increases in ammonia-nitrogen and nitrite concentrations in the water caused by fish metabolites during transportation is often considered as one of the important factors causing transport stress [1] and increased mortality in the fish after transport [2]. Transport stress can cause irregular oxidation in the aerobic metabolic pathway, resulting in oxidative stress in transported fish [3,4]. Mechanical damage caused by turbulence and congestion during transport can lead to immune injury, making fish susceptible to pathogenic infection [5]. Transport stress significantly affected glycogen, superoxide dismutase (SOD) and malondialdehyde (MDA) contents in hybrid snapper (*Pagrus major*♀ × *Acanthopagrus schlegelii*♂) livers, with significantly increased mortality in these fish after transport [6]. One study found that liver lysozyme and IgM activity increased significantly in hybrid *Pelteobagrus fulvidraco* (*Tachysurus fulvidraco*♀ × *Pseudobagrus vachellii*♂) after transport stress [7]. Another study found that alkaline phosphatase, acid phosphatase, SOD activity and total antioxidant capacity (T-AOC) in the gills first increased, then decreased [8]. Additionally, short-distance transport stress significantly decreased the SOD and T-AOC activities in liver tissue from *Oncorhynchus mykiss* [9]. T-AOC, CAT and MDA levels first increased, then decreased during transport in *Ictalurus punctatus* [2], and T-AOC, lysozyme and complement C3 activities increased significantly in blunt snout bream (*Megalobrama amblycephala*) after transport [10].

The intestines are important for digestion and absorption in fish, and the gut microbiome participates in metabolism and synthesis of proteins, amino acids and other substances [11], which is important in physiological metabolism and immunity. Gut microbiome stability is important for maintaining host health [12]. Environmental stress can change the gut microbiome structure [13]. Crowding stress significantly changed the gut microbial abundances in blunt snout bream (*Megalobrama amblycephala*) at the genus level, with a significant correlation between intestinal microorganisms and 13 metabolites [14]. In *Penaeus vannamei*, stress from high ammonia-nitrogen concentrations significantly decreased the gut microbial abundances and reshaped the genus-level community structure of the gut microbiome [15]. Additionally, transport stress changed the intestinal microbial diversity of hybrid yellow catfish (*Tachysurus fulvidraco*♀ × *Pseudobagrus vachellii*♂) and affected host microbial functions [16].

Largemouth bass (*Micropterus salmoides*) are native to freshwater basins in North America and are now widely farmed throughout China. This species is economically important owing to its fast growth and strong disease resistance [17,18]. However, breeding largemouth bass seedlings is concentrated in Jiangsu and Guangdong, and the fingerling often need to be transported long distances, which can lead to decreased vitality and sometimes death, which adversely affects healthy breeding and limits rapid development in the aquaculture industry. This study was conducted to observe the effects of transport stress on liver and intestinal tissue structures in largemouth bass by simulating long-distance transport. We also analyzed the effects of transport stress on antioxidant and immune abilities in largemouth bass and clarified the differences in gut microbiome compositions before and after transport stress. The results provide a reference for optimizing transportation of juvenile largemouth bass.

## 2. Materials and Methods

### 2.1. Experimental Materials

Largemouth bass (50 days of age with average body length 8.42 ± 0.44 cm, average weight 10.26 ± 0.32 g) were selected from the Yixing Base of Freshwater Fisheries Center, Chinese Academy of Fishery Sciences (Wuxi, China). The bass were cultured in a recirculating aquaculture system consist of 26 cylindrical circulation barrels with a diameter of 1.0 m, height of 1.2 m, and the water used for aquaculture was filtered pond water. The bass were fed commercial feed (crude protein 46.0% and crude fat 6%) at 2% of their body weight twice daily (8:00 am and 16:00 pm) in a recirculating aquaculture system with density of 1.3 g/L at 24.0 ± 0.3 °C, ammonia-nitrogen <0.01 mg/L, and dissolved oxygen >6 mg/L before the experiment. The culture conditions were applied to the control and recovery groups. The fish was fasted 24 h immediately after last feeding, then used for experiment.

### 2.2. Experimental Design

The experiment was divided into treatment groups: 0-h, 6-h, and 12 h transport stress; 5 d recovery after a 12 h simulated transport, and a control group. Three parallels were set per group. The bass that remained in the recirculating aquaculture system were used as controls. Twelve double-layered plastic bags (40 × 80 cm) containing one-third water and two-third oxygen were placed on an automated shaker (Mince instrument, Changzhou, China) to simulate the actual transport. The vibration frequency was set at 100 rpm [6,16], and each bag contained 15 fish. The air conditioning temperature was set to 22 degrees to ensure that the ring temperature was constant during the simulated transportation, and avoided all light in the process. Nine bags were used for sampling at 0, 6, and 12 h of transportation. The other three bags were placed in the recirculating aquaculture system for recovery after 12 h of transport stress, then sampled after 5 d of recovery. The recovery conditions were the same with the control group.

### 2.3. Sample Collection

Three bags were randomly selected for sampling at each time point, water samples were taken, 15 fish were randomly selected and 5 of the 15 fish were sampled from each of the 3 bags. Control water and fish were obtained from the recirculating aquaculture system at the corresponding time points. The fish were anesthetized via 200 mg/L MS-222 before sampling. The livers and posterior section of the intestines were collected from three fish and fixed with 4% paraformaldehyde solution for histological analysis. The livers and intestines from the remaining 12 fish were homogenized and mixed with precooled phosphate-buffered saline, then centrifuged for 15 min at 12,000× *g* at 4 °C. The supernatant was aspirated and stored at −80 °C for physiological parameter analysis. The hindguts were collected from five fish each from the control, 12 h transport and recovery groups, immediately frozen in liquid nitrogen, and stored in a −80 °C freezer for gut microbiome analysis.

### 2.4. Water Quality Detection

The dissolved oxygen was measured using an oxygen-dissolving meter (Hach, Loveland, CO, USA), and the total ammonia-nitrogen and nitrite-nitrogen contents were determined via spectrophotometry [7].

### 2.5. Histological Analysis of the Liver and Intestinal Tissues

The tissues were immersed in 4% paraformaldehyde for 24 h, then routinely processed, embedded in paraffin, and sectioned. The sections were dewaxed in xylene for 2–5 min, then washed continuously in 100%, 96%, 80%, and 70% ethanol for 1 min. Sections were then stained with hematoxylin for 7 min, rinsed with distilled water for 2 min, rinsed with 0.1% hydrochloric acid and 50% ethanol for 2–5 s, rinsed with tap water for 5–7 min, stained with eosin for 2–4 min, rinsed with distilled water for 1 min, dehydrated with 95% and 100% ethanol for 1 min each and finally rinsed with xylene (2–5 min) [19]. The sections were then air-dried, and the slides were covered and observed under a light microscope. Hematoxylin stained the normal nuclei blue. The liver tissue size and cavitation ratio (cavitation ratio [%] = cell vacuolar area/section area×100) were measured using Image-Pro Plus 6.0 [20].

### 2.6. Liver and Intestinal Biochemical Analyses

Liver SOD, MDA, lysozyme, complement C3, intestinal SOD and MDA activities were detected using commercial kits (Jiancheng Institute, Nanjing, China) following the manufacturer’s instructions [16].

### 2.7. Determination of the Gut Microbiome

The E.Z.N.A.^®^ Stool DNA Kit (D4015, Omega Inc., Norcross, GA, USA) was used to extract microbial DNA from the intestinal samples, and the DNA sample quality was detected via 1% agarose gel electrophoresis and quantified using an ultraviolet spectrophotometer. Samples with correct target bands were regarded as qualified samples. PCR was performed using universal primers: v3-v4 region [21]: 341F: 5′-CCTACGGGNGGCWGCAG-3′ and 805R: 5′-GACTACHVGGGTATCTAATCC-3′. The PCR reaction system consisted of 50 ng template DNA, 12.5 µL of PCR Premix, 2.5 µL each of forward and reverse primers and ddH2O added to 25 µL. The PCR amplification reaction was 98 °C predenaturation for 30 s, 35 cycles of 98 °C denaturation for 10 s, 54 °C annealing for 30 s, and a 72 °C extension for 45 s. Extension was continued at 72 °C for 10 min at the end of the cycle, and finally, preserved at 4 °C. The product quality was detected via 1% agarose gel electrophoresis. Correctly sized target bands with 700 bp were considered qualified samples. PCR products were purified using AMPure XT beads (Beckman Coulter Genomics, Danvers, MA, USA) and quantified using Qubit (Invitrogen, Carlsbad, CA, USA). The sequencing libraries were prepared by Pacific Biosciences SMRTbell^TM^ Template Prep kit 1.0 (Kapa Biosciences, Woburn, MA, USA). The size and number of amplicon libraries were assessed using a library quantification kit from Illumina (Kapa Biosciences, Woburn, MA, USA) and an Agilent 2100 Bioanalyzer (Agilent, Santa Clara, CA, USA).

Eligible libraries were sequenced on the Illumina NovaSeq platform and read by FLASH merge matching ends. Under specific filtering conditions, fqtrim (v0.94) was used to quality-filter the raw read data to obtain high-quality clean labels. Chimeric sequences were filtered with the Vsearch (v2.3.4) software. The feature table and sequence were obtained by demodulation using DADA2. Observed species, Chao1, Shannon and Simpson indices were used to evaluate alpha diversity. Beta-diversity indexes were used to evaluate species composition differences among samples. The alpha and beta diversity were calculated using QIIME2. Species with significantly different abundances between groups were analyzed with the nonparametric factor Kruskal–Wallis rank-sum test. The PICRUSt2 software was used for Kyoto Encyclopedia of Genes and Genomes (KEGG) enrichment analysis.

### 2.8. Statistical Analysis

Data were analyzed and processed using SPSS 26.0 (SPSS Inc., Chicago, IL, USA). Shapiro–Wilk and Levene tests were used to analyze the normality and variance homogeneity of the data, and single factor analysis of variance (one-way ANOVA) was used to analyze significant differences among groups by Duncan’s multiple comparisons. The control and treatment groups were compared using independent sample t-tests. The Kruskal–Wallis method was used to analyze the significant differences in alpha diversity indexes among samples, and correlations between the gut microbiome and physiological parameters were analyzed via Spearman’s correlation analyses. Box plots were drawn using origin software; all other pictures were constructed in R [22]. *p* < 0.05 was considered significant, and all data were expressed as means ± standard error of the mean.

## 3. Results

### 3.1. Water Quality during Transport

The total ammonia-nitrogen and nitrite-nitrogen concentrations increased significantly as the transportation time increased (*p* < 0.05, Figure 1A,B). After 12 h of transportation, the total ammonia-nitrogen concentration reached 0.723 ± 0.009 mg/L, which was significantly higher than that of its control group (0.0163 ± 0.002 mg/L), and the nitrite-nitrogen concentration was 0.129 ± 0.008 mg/L, which was significantly higher than that of its control group (0.009 ± 0.002 mg/L). The dissolved oxygen was kept above 20 mg/L during transport (Figure 1C).

### 3.2. Histological Analysis of the Liver and Intestinal Tissues

After 0 h of transport stress, liver cells from the juvenile largemouth bass were undamaged, and the cells exhibited obvious boundaries (Figure 2(Aa)). After 6 h of transport stress, vacuoles appeared in the liver cells, at proportions reaching 22.60% (Figure 2(Ab); Table 1). After 12 h of transport stress, the liver cells showed severe vacuolization (Figure 2(Ac)), with proportions reaching 60.34%, which was significantly higher than that after 6 h of transport stress (*p* < 0.01), and the healthy hepatocytes were squeezed as the vacuoles expanded. After 5 days of recovery, the vacuolar area remained significantly larger than that of the control group (37.06%; *p* < 0.05; Figure 2(Ad)).

Hematoxylin-eosin staining showed that the intestinal muscle layer thickness increased and the intestinal villus length decreased under transport stress (Figure 2B). The intestinal muscle layer thickness did not significantly differ among groups at 0, 6 and 12 h of transport (*p* > 0.05). However, the thickness in the recovery group was significantly greater than those at 0 and 6 h of transport stress (*p* < 0.05) and did not significantly differ from that at 12 h of transport stress (*p* > 0.05). The villus length decreased as the transport time increased and remained significantly shorter after 5 days of recovery (*p* < 0.05; Table 1). No fish died during transport.

### 3.3. Oxidative and Immunological Parameters

None of the measured physiological parameters differed significantly in the control group (Figure 2). However, SOD activity in the livers and intestines of the juvenile largemouth bass increased significantly as the transport time increased (*p* < 0.05), and the SOD activity peaked after 5 days of recovery, which was significantly higher than that after 12 h of transport stress (Figure 3A,E; *p* < 0.05). MDA concentrations in the livers and intestines peaked after 6 h of transportation, decreased significantly after 12 h of transport (*p* < 0.05), then increased significantly after 5 days of recovery (Figure 3B,F; *p* < 0.05). Lysozyme activity increased as the transport stress time increased (*p* < 0.05), with the highest activity at 12 h of transport stress. No significant difference was noted in the recovery group (Figure 3C). Complement C3 levels increased rapidly from 6 to 12 h of transport stress (Figure 3D; *p* < 0.05), with no significant difference between the 12 h transport group and the recovery group. The SOD, MDA, lysozyme and complement C3 levels differed markedly between the recovery and control groups.

### 3.4. Transport Stress Effects on the Gut Microbiome 

To investigate the effects of transport stress and recovery on the gut microbiomes of juvenile largemouth bass, the gut microbiomes of the control, 12 h transport stress and 5 d recovery groups were analyzed. High-throughput sequencing yielded 3207 characteristic data points of which 157 were shared by the three groups. The recovery group had the least characteristic data (Figure 4A). Alpha diversity indexes (i.e., observed species, Shannon, Simpson, and Chao1 indexes) were used to assess the richness and evenness of the intestinal microbiotas (Figure 4B), which did not significantly differ among the groups (*p* > 0.05).

Principal coordinates analysis based on weighted and unweighted UniFrac distances was used to describe the beta diversity of the intestinal flora to quantify the microbial community compositions in each group (Figure 4C). The samples in each group were clustered together. The distance between the control and treatment groups was long, and the overlap between the 12 h transport and recovery groups was large. 

The gut microbiome compositions of the juvenile largemouth bass were analyzed at the phylum and genus levels. At the phylum level (Figure 5A), Proteobacteria, Firmicutes, Cyanobacteria and Spirochaetes were predominant. The box plots describe their abundance changes (Figure 5B). The Proteobacteria abundance increased significantly after transport stress (*p* < 0.05), with no significant difference between the 12 h transport and recovery groups. Firmicutes increased in abundance after transport stress, but did not significantly differ among the groups. The relative abundances of Cyanobacteria and Spirochetes were higher in the control group, but decreased significantly after 12 h of transport stress (*p* < 0.05). The abundance was maintained in the recovery group.

Linear discriminant analysis results (Figure 6A,B) showed significant differences in the abundances of *Plesiomonas*, *Cyanobium_PCC_6307*, *Firmicutes_unclassified*, *Brevinema Clostridium_sensu_stricto_1* and *Cetobacterium*. The *Plesiomonas* abundance increased significantly after transport stress, and was maintained in the recovery group (Figure 6(Ca)). The relative abundances of *Cyanobium_PCC_6307* and *Brevinema* were significantly higher in the control group than in the 12 h transport stress and 5 d recovery groups (Figure 6(Cb,Cd); *p* < 0.05). The relative abundances of *Clostridium_sensu_stricto_1* and *Firmicutes_unclassified* were significantly higher in the 5 d recovery group than in the control and 12 h transport groups (Figure 6(Cc,Cf); *p* < 0.05). The relative abundance of *Cetobacterium* was significantly higher in the 12 h transport group than in the control group but was similar to that of the recovery group (Figure 6(Ce)).

Bacteria with significant changes in abundance were used to analyze the correlations with intestinal SOD, MDA, villus length and muscle layer thickness. The measured intestinal parameters were strongly correlated with the gut microbiome at the genus level (Figure 7A). The MDA content was correlated with all bacterial genera, the intestinal villus length and muscle layer thickness were correlated with five genera, and the SOD content was correlated with four genera.

KEGG functional analysis showed that that gut microbiome was mostly enriched in membrane transport, replication and repair, and translation (Figure 7B). Compared with the control group, fewer gut microbes were involved in replication and repair and translation, and more functional bacteria were involved in membrane transport in the 12 h transport and 5 d recovery groups.

## 4. Discussion

### 4.1. Water Quality Parameters during Transport Stress

Many factors, such as water quality changes, fish body bruising, starvation and so on, may cause transport stress during fish transportation. The total ammonia-nitrogen, nitrite-nitrogen and oxygen concentrations are the important factors causing transport stress in fish [1]. Studies have shown that increased ammonia-nitrogen concentrations can destroy the osmotic pressure balance and cause oxidative stress [3], while increased nitrite-nitrogen concentrations can poison fish and inhibit nonspecific immunity [23]. In this experiment, while the ammonia-nitrogen and nitrite-nitrogen concentrations increased significantly (ammonia-nitrogen concentrations from 0.01 ± 0.00 mg/L to 0.72 ± 0.09 mg/L and nitrite-nitrogen concentrations from 0.01 ± 0.00 mg/L to 0.13 ± 0.02 mg/L) as the transport time increased, the detected N levels were still ~20-fold below those considered to be harmful to wild fish [24]. Even so, elevated aquatic N levels may have increased the transport stress incurred by the largemouth bass, possibly owing to increases in fish metabolites. This variation tendency is consistent with previous studies [6,7]. Increased ammonia-nitrogen and nitrite concentrations during transport stress have been shown to cause oxidative stress responses in channel catfish and nonspecifically damage hybrid yellow catfish [4,7]. This may be due to the increased ammonia-nitrite concentration accelerating reactive oxygen species (ROS) production, thus causing oxidative stress during transport. Additionally, deteriorated water quality can cause pathogen proliferation and damage nonspecific immunity among largemouth bass. In our study, the oxygen concentration (about 20 mg/L) remained relatively stable but four times higher than the control group under transport stress, possibly because the transport bags contained two-thirds oxygen, which caused hyperoxia to fish. Rainbow trout exposed to intermittent hyperoxia showed significant oxidative stress [25]. Hyperoxia results in transient oxidative stress and alters antioxidant enzymes in goldfish [26]. Tristan et al. [27] believe that hyperoxia can affect the level of ROS in fish and give rise to oxidative stress. So, hyperoxia may be one of the reasons for the changes in the antioxidant enzyme activities in this study.

### 4.2. Effects of Transport Stress on Liver and Intestinal Structure in Juvenile Largemouth Bass

The liver is involved in the synthesis and decomposition of various substances in fish. Vacuolation, edema and necrosis of liver cells are all signs of damage to fish livers under environmental stress [28]. Safahieh et al. [29] believe that liver cell vacuolization indicates degeneration before cell necrosis. In this study, as the transport stress increased, the liver cell volume and number of vacuoles increased, possibly related to energy metabolism increases in largemouth bass to maintain homeostasis during transport, resulting in an altered liver glycogen content, leading to liver cell vacuolization [30]. Severe vacuolization in the liver tissue persisted even after the 5-day recovery, indicating that the liver damage caused by transport stress could not be recovered within 5 days.

Villus length, villus width and muscle thickness enable evaluating the intestinal health of fish [31]. In this experiment, the intestinal villus length was significantly shortened after transport stress, suggesting intestinal damage and likely affected nutrient digestion and absorption in juvenile largemouth bass [32]. Additionally, transport stress significantly increased the intestinal muscle layer thickness, possibly because it induced intestinal inflammation. Studies have shown that the shortened intestinal villus length and altered muscle layer thickness may be related to intestinal inflammation [31,33].

### 4.3. Effects of Transport Stress and Recovery on Antioxidant and Immune Enzyme Activities in Juvenile Largemouth Bass

SOD is cells’ first-line defense against toxic free radicals can remove excessive ROS in fish [34]. In this study, the SOD activity increased significantly under transport stress, likely due to the increased ammonia-nitrogen and nitrite concentrations and the long crowding conditions, leading to excessive ROS production in transported fish [35]. Meanwhile, the hyperoxia environment also could be the underlying cause for enzyme activity changes or increased levels of ROS [27]. ROS are harmful substances produced by metabolic activities in fish, which can cause oxidative stress. Increased SOD activity accelerates ROS removal. Additionally, increased activity of SOD may be related to the accelerated generation of ROS. A study on hybrid snapper also showed that SOD activity increases under transport stress [6]. Free radicals and ROS can damage fish livers [36] by damaging the liver cell biofilm through lipid peroxidation (LPO), resulting in liver cell damage [37]. Oxidative stress caused by transport stress may have been one reason for the liver damage in this study. Biofilm oxidation by ROS leads to LPO, and MDA, the final product of LPO, is an important indicator of antioxidant capacity in fish [6]. Previous studies have shown that oxidative stress caused by transport stress significantly increased MDA contents in fish [2,6], and the significant increase in MDA content after 6 h of transport stress indicated oxidative stress caused by transportation. However, the MDA content decreased significantly in the 12 h transport group, possibly owing to accelerated MDA metabolism under high-oxygen conditions (dissolved oxygen >20 mg/L). Fish can regulate enzymatic and nonenzymatic antioxidant defense systems to accelerate MDA clearance under hyperoxic conditions [38]. Additionally, oxidative stress can reduce polyunsaturated fatty acid contents by damaging cell membrane structures, which can be oxidized and produce MDA, thus reducing MDA concentrations [39].

The lysozyme activity and complement C3 concentration increased significantly after transport stress. Lysozyme is an important component of fish’s nonspecific immunity and can lyse bacterial cell walls and effectively resist pathogen invasion [9,40]. Complement comprises the plasma protein family and is the core component of innate immunity [41]. Complement C3 helps fish recognize invading microorganisms, mark damaged host cells, and help phagocytes eliminate pathogenic bacteria and damaged cells [42]. Under environmental stress, fish activate homeostatic regulatory mechanisms to affect the body’s immune response [43]. The increased lysozyme and complement C3 activity indicate an enhanced immune response in fish. Transport stress experiments with *Pelteobagrus fulvidraco* and *Piaractus mesopotamicus* showed increased lysozyme and complement C3 contents [7,44], possibly because the transport stress impaired part of the immune capacity of the largemouth bass, combined with the deteriorated water quality accelerating pathogen proliferation, which exacerbated bacterial infection in the fish, thus activating the lysozyme and complement C3.

The lysozyme and complement C3 activities were significantly higher in the recovery group than in the 0 h transport group. Studies on transport stress in *Lateolabrax maculatus* and *Piaractus mesopotamicus* have also reported this phenomenon [42,43], possibly due to the susceptibility of the fish after transport stress, sustaining the immune response. There is a study that shows that transport stress can damage the immune system of transported fish [44], making them vulnerable to bacterial infection. Studies have shown that pathogen infection often leads to excessive ROS production in animals [45], resulting in oxidative stress, which may be why the SOD activity in the recovery group was significantly higher than that of the 12 h transport stress without recovery. Therefore, we suspect that the infectibility of the largemouth bass after transportation caused their inability to recover.

### 4.4. Effects of Transport Stress on Gut Microbial Diversity and Structure in Juvenile Largemouth Bass

In this experiment, alpha diversity did not significantly differ among the groups, possibly owing to individual factors in largemouth bass. Yan et al. [46] suggested that the influence of the fish itself on its gut microbiome was much greater than that of the environment. Sullam et al. [47] studied *Poecilia reticulata* and proposed that its core flora was strongly correlated with the host genotype, which does not easily fluctuate when affected by the external environment. Largemouth bass are carnivorous; their gut microbiome may be stable even with different treatments, and no significant differences were noted in their gut microbiome alpha diversity [48,49]. Therefore, largemouth bass gut microbiomes may be associated with their own genes, and their alpha diversity is not significantly altered under transport stress. However, significant differences were observed in beta diversity, suggesting that transport stress did not alter the gut microbial richness or evenness but significantly affected the gut microbial composition and distribution in juvenile largemouth bass. Changing community structures can easily cause abnormal host physiological functions [50], thus affecting nutrient digestion and absorption [51]. Additionally, correlation analysis results showed a significant correlation between gut microbes at the genus level and the measured intestinal parameters; however, the connection between these bacteria and intestinal parameters during transport stress requires further study.

Gut microbes perform diverse functions within the host’s gut and are closely related to host health [50]. Proteobacteria are the largest branch of prokaryotes and include pathogens such as *Escherichia coli*, *Salmonella*, *Vibrio*, and *Helicobacter pylori* [52]. Increased numbers of Proteobacteria are a potential marker of fish infection [53]. In this study, the Proteobacteria abundance increased markedly after transport stress, which may have increased the susceptibility to infectious diseases in juvenile largemouth bass. *Plesiomonas*, the dominant genus in this study, is a Gram-negative bacterium of the Enterobacteriaceae family. *Shigella* is also related to outbreaks of fish disease [54]. We found that the abundance of *Plesiomonas* in the 12 h transport group was significantly higher than that of the control group, which might indicate the infectivity of juvenile largemouth bass after transport stress. Therefore, juvenile largemouth bass may be more susceptible to disease after long-distance transportation. The high abundance of Proteobacteria in the 5 d recovery group showed that largemouth bass remained susceptible to bacteria after 5 days, which is consistent with the results for the antioxidant and immune enzyme activities. KEGG functional analysis showed that few gut microbes were enriched in metabolic diseases; therefore, we suspect that increased abundances of Proteobacteria may be a marker of susceptibility to infection in juvenile largemouth bass and that Proteobacteria are not directly pathogenic to largemouth bass.

Firmicutes play roles in physiological processes such as polysaccharide degradation [52,55]. Smriga et al. [56] found that Firmicutes played a role in host digestion. Studies have shown that Firmicutes are related to lipid metabolism in animals [57] and can accelerate the metabolism of the precursor of MDA: total polyunsaturated fatty acids, which might decrease MDA contents. Firmicutes abundances did not significantly differ among groups. However, the abundances of *Clostridium_sensu_stricto_1* and *Firmicutes_unclassified* were significantly increased under transport stress. Therefore, transport stress likely significantly affected only some Firmicutes. *Clostridium* can enhance glucose metabolism, promote fish growth [58], and increase energy metabolism levels in fish. Changes in the ammonia-nitrogen concentration between transportation and recovery can activate osmotic mechanisms and activate osmotic regulation [59]. KEGG functional analysis also showed that intestinal microorganisms were mainly enriched in membrane transport, which may enhance energy metabolism after transport. This may have increased the abundances of *Clostridium_sensu_stricto_1* and *Firmicutes_unclassified*. When more energy is used to defend against pathogens and regulate the osmotic balance, glucose metabolism occurs via glycolysis under hormone regulation, resulting in large amounts of liver glycogen being stored in the liver and may, therefore, lead to liver cell swelling and vacuolization [28,60].

In this study, the Spirochetes abundance was significantly higher in the control group than in the 12 h transport and 5 d recovery groups. Spirochetes are endosymbionts with some arthropods and mollusks and participate in lignocellulose decomposition and nitrogen fixation in termite intestines [61]. Spirochetes may play roles in host nutrient metabolism [62], and decreased Spirochetes abundances may indicate physiological damage in largemouth bass. Cyanophyta is a primitive green autotrophic plant group, and its physiological functions remain unclear. It can pass into the gut combined with feed, and showed high abundance in the control group. It may be passed out of the gut with the increase in fasting time. Whether Cyanophyta plays a role in antioxidant and physiological immune functions in largemouth bass requires further study. It is worth noting that Cyanophyta abundance was markedly lower in the 5 d recovery groups than in the control group, which may arise from a vitality decrease and eating less after transport stress.

## 5. Conclusions

Deterioration of water quality caused by transport stress can cause oxidative stress and activate immune responses in largemouth bass. Changes in the intestinal microbial community revealed that the gut microbiotas of largemouth bass are involved in adaptation to transport stress. These variations manifested as changes in the liver and intestinal structure. Additionally, juvenile largemouth bass required more than 5 days to recover after 12 h of transportation due to increased infection susceptibility. These results may provide a theoretical basis and support for clarifying the transport stress-induced oxidative and immune mechanisms of juvenile largemouth bass.

## Figures and Tables

**Figure 1 antioxidants-12-00157-f001:**
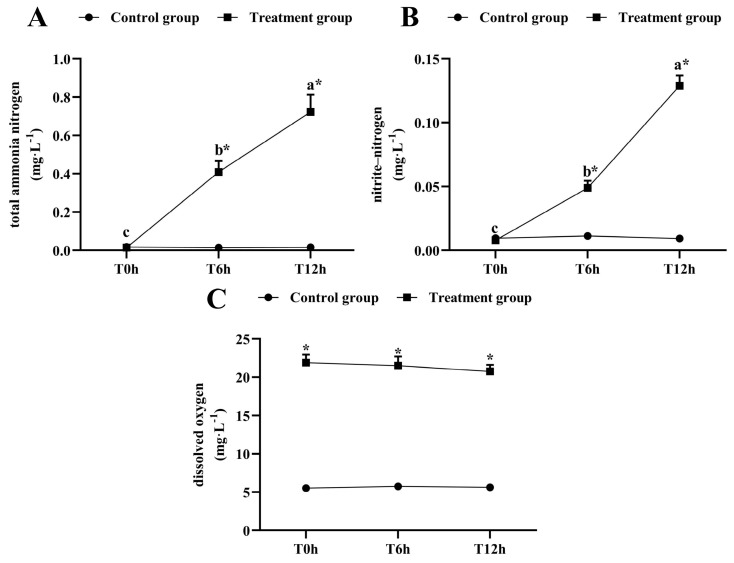
Changes in total ammonia-nitrogen (**A**) nitrite-nitrogen (**B**) and dissolved oxygen (**C**) levels during transport. T0h: transport stress at 0 h; T6h: transport stress at 6 h; T12h: transport stress at 12 h. * significant difference between control and treatment groups (paired samples *t*-test, *: *p* < 0.05). Lowercase letters indicate significant differences (*p* < 0.05) between treatments (Duncan’s multiple range test).

**Figure 2 antioxidants-12-00157-f002:**
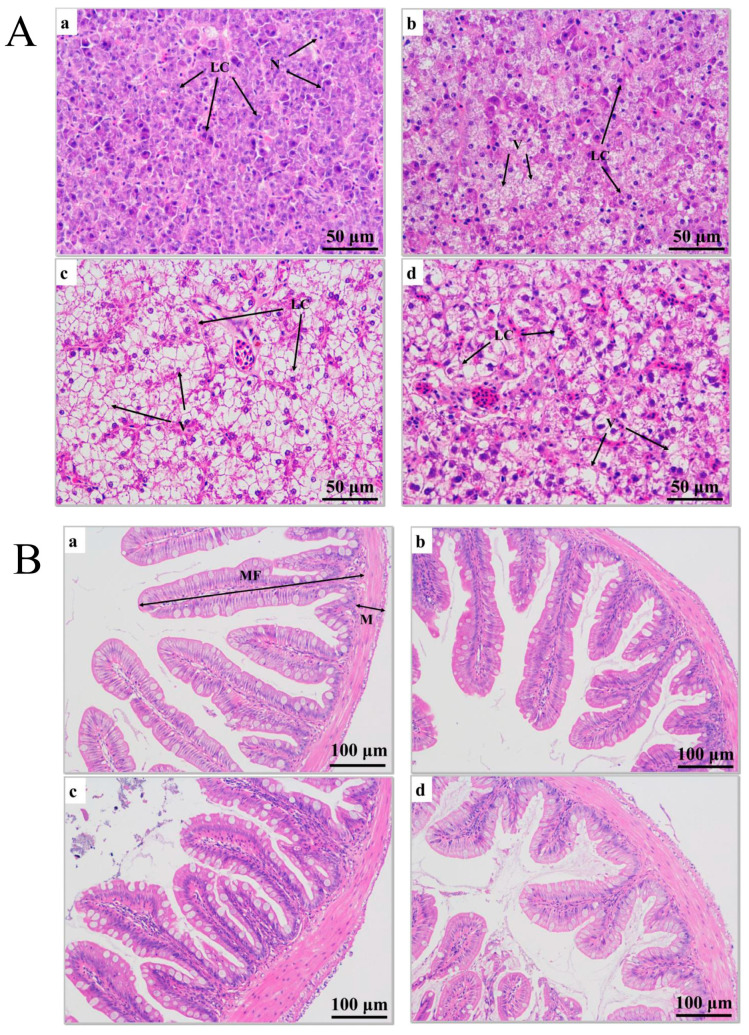
(**A**) Histopathological sections of juvenile largemouth bass livers under transport stress. LC: liver cells; N: nucleus; V: vacuoles. Scale 1:400. (**B**) Histopathological sections of juvenile largemouth bass intestines under transport stress. MF: intestinal villi; M: muscular layer. Scale 1:200. Sections under transport stress at 0 h (**a**), 6 h (**b**), 12 h (**c**) and after 5 d of recovery (**d**) after transportation.

**Figure 3 antioxidants-12-00157-f003:**
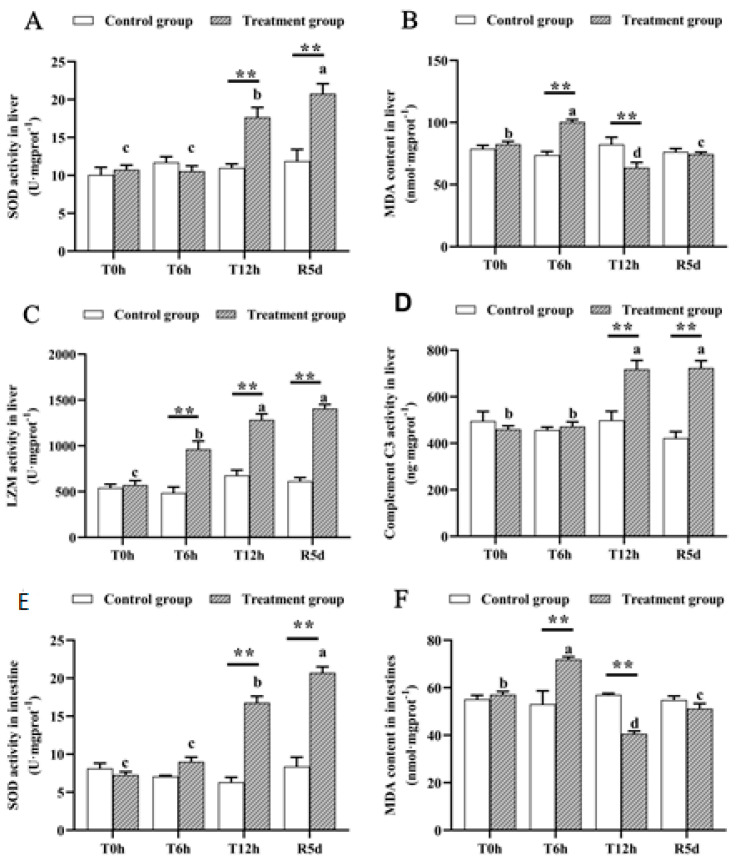
Antioxidant- and immune-related physiological parameters in the livers and intestines of juvenile largemouth bass under transport stress. Superoxide dismutase, SOD; malondialdehyde, MDA; lysozyme, LZM (**A**) liver SOD; (**B**) liver MDA; (**C**) liver LZM; (**D**) liver complement C3; (**E**) intestinal SOD; (**F**) intestinal MDA. T0h: transport stress at 0 h; T6h: transport stress at 6 h; T12h: transport stress at 12 h; R5d: recovery for 5 d. **: significant difference between the control and treatment groups (paired samples *t*-test, **: *p* < 0.01). Lowercase letters indicate significant differences (*p* < 0.05) between treatments (Duncan’s multiple-range test).

**Figure 4 antioxidants-12-00157-f004:**
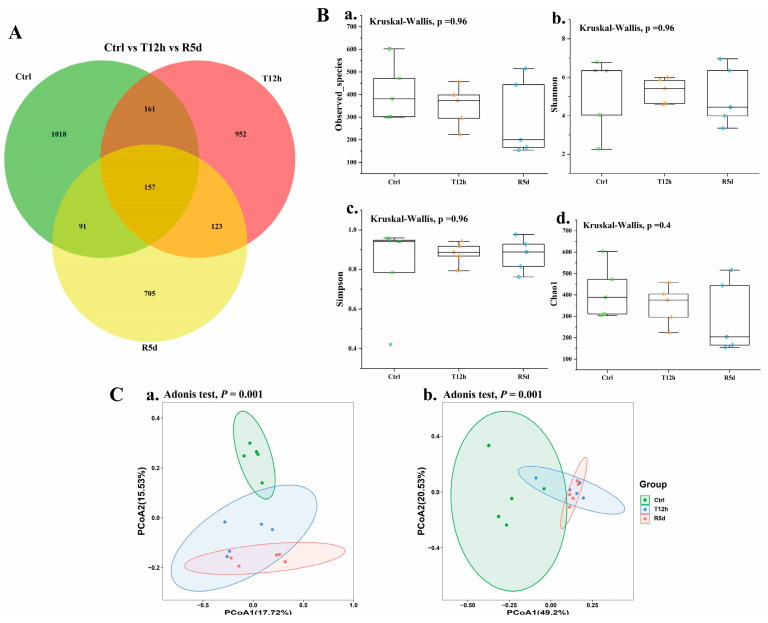
(**A**) Common and unique operational taxonomic units (OTUs) displayed by Venn diagram in the control, 12 h transport and recovery groups. The overlap indicates shared OTUs (**B**) and alpha and beta diversity of the gut microbiomes of juvenile largemouth bass under transport stress. Measures of alpha diversity included: (**a**) Observed species; (**b**) Shannon index; (**c**) Simpson index; (**d**) Chao1 index. (**C**) Beta diversity analysis based on principal coordinates analysis of unweighted UniFrac (**a**) and weighted UniFrac (**b**) distances. Ctrl: control group; T12h: transport stress for 12 h; R5d: recovery for 5 d.

**Figure 5 antioxidants-12-00157-f005:**
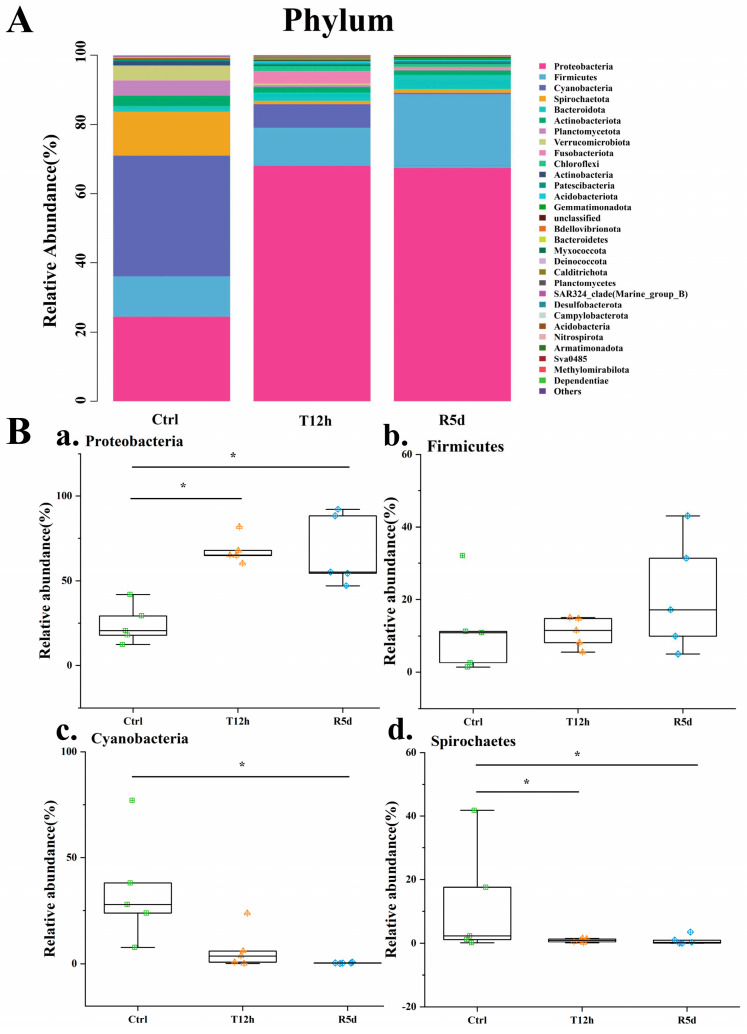
(**A**) Relative abundances of dominant phylum-level taxa in the control (Ctrl), 12 h transport (T12h) and recovery (R5d) groups. Species with lower abundances were classified as “others”. (**B**) Relative abundances of the dominant phylum-level bacteria (**a**–**d**) in the Ctrl, T12h and R5d groups. * significant difference between the control and treatment groups (Kruskal-Wallis H test, *: *p* < 0.05).

**Figure 6 antioxidants-12-00157-f006:**
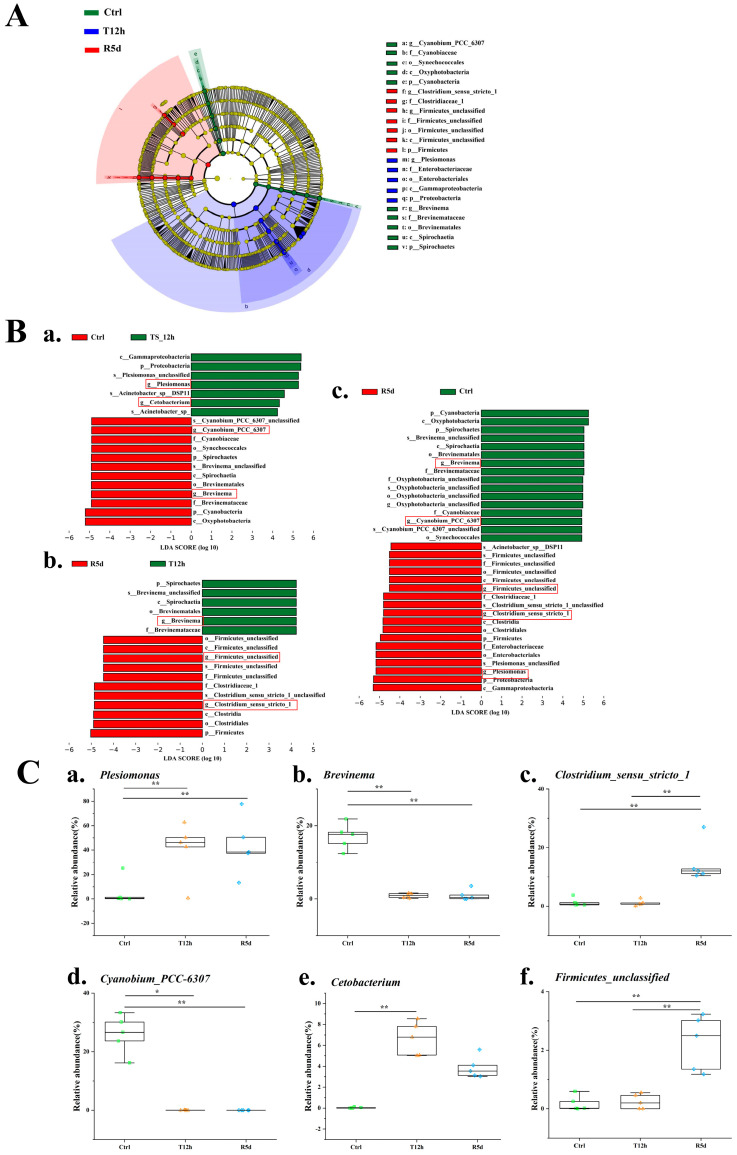
Linear discriminant analysis effect size (LEfSe) analysis comparing abundances of bacterial taxa in the control (Ctrl), 12 h transport (T12h) and recovery (R5d) groups. The LDA score >4. (**A**) Cladogram based on LEfSe analysis. Green, blue and red represent taxa enriched in the Ctrl, T12h and R5d groups. (**B**) Bacterial taxa with different abundances among groups. (**a**) Ctrl vs. T12h, (**b**) Ctrl vs. R5d, (**c**) T12h vs. R5, and (**d**) by Kruskal–Wallis test. (**C**) Relative abundances of dominant bacteria at the genus level (**a**–**f**) in the Ctrl, T12h and R5d groups. *, **: significant differences between the control and treatment groups (Kruskal–Wallis H test, *: *p* < 0.05, **: *p* < 0.01).

**Figure 7 antioxidants-12-00157-f007:**
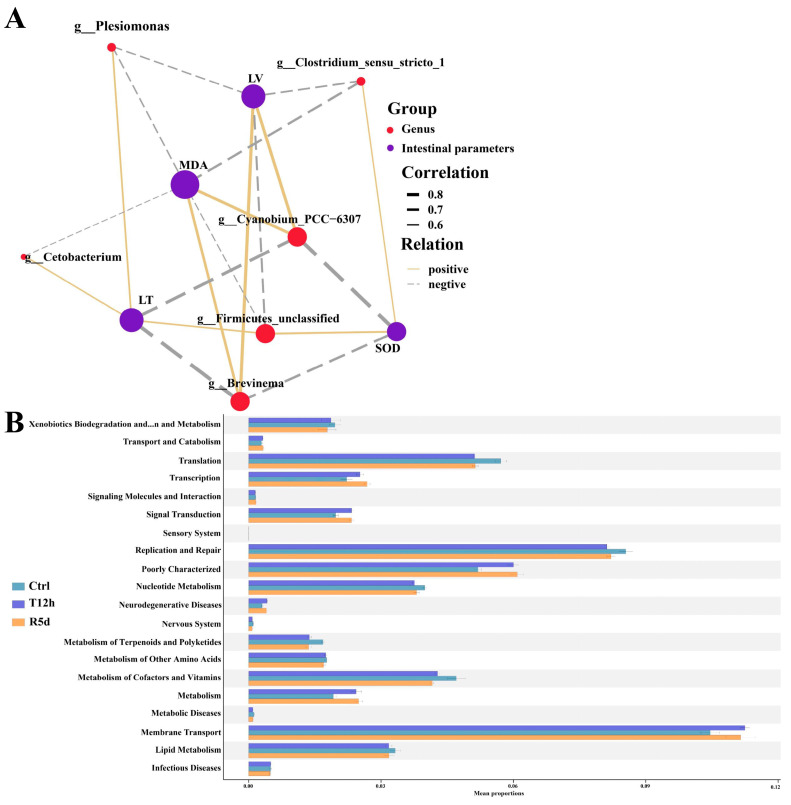
(**A**) Correlation network showing connections between gut microbial genera and intestinal parameters. Line thickness indicates the magnitude of the correlation. LV: intestinal villus length; LT: muscular thickness. (**B**) Functional analysis of the gut microbiota predicted at Level 2 for the control, 12 h transport and recovery groups.

**Table 1 antioxidants-12-00157-t001:** Changes in liver vacuolar area, intestinal muscular thickness and villus length in juvenile largemouth bass under transport stress.

Items	Groups							
	T0h	Ctrl	T6h	Ctrl	T12h	Ctrl	R5d	Ctrl
Liver vacuolar area (%)	0.81 ± 0.02 ^d^	0.37 ± 0.04	23 ± 0 ^c^	1.04 ± 0.10	60 ± 1 ^a^	0.97 ± 0.07	37 ± 1 ^b^	1.36 ± 0.47
Muscularis thickness (μm)	39 ± 5 ^b^	37 ± 4	39 ± 3 ^b^	40 ± 2	50 ± 7 ^ab^	34 ± 4	56 ± 2 ^a^	37 ± 8
The length of intestinal villus (μm)	391 ± 7 ^a^	378 ± 10	316 ± 21 ^b^	415 ± 18	272 ± 16 ^c^	420 ± 13	193 ± 11 ^d^	381 ± 4

T0h: transport stress at 0 h; T6h: transport stress at 6 h; T12h: transport stress at 12 h; R5d: 5-day recovery after transport stress; Ctrl: control group at the corresponding time points. Lowercase letters indicate significance differences (*p* < 0.05).

## Data Availability

The 16S rRNA datasets are under accession number PRJNA853451. The raw data supporting the conclusions of this article will be made available by the authors without undue reservation. Requests to access these datasets should be directed to qiangj@ffrc.cn.

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
