# Peer review of "Transport Stress Induces Oxidative Stress and Immune Response in Juvenile Largemouth Bass (Micropterus salmoides): Analysis of Oxidative and Immunological Parameters and the Gut Microbiome"

_antioxidants, 2023, doi:10.3390/antiox12010157_

Round 1
Reviewer 1 Report
Transport stress induces oxidative stress and immune response in juvenile largemouth bass (Micropterus salmoides):analysis of oxidative and immunological parameters and the gut microbiome
Wang Q, Ye W, Tao Y, Li Y, Lu S, Xu P, Qiang J
In this article, the authors describe the stress responses caused by transport using oxidative stress and immunological parameters, histological parameters, and gut microbiome in juvenile largemouth bass. The changes in the parameters are clearly stated. In addition, a dynamic of bacterial species in the gut is discussed in relation to the functional aspects of the bacterial species. The experiment was well designed and well done. The review gives a latest knowledge about the stress response of the fish caused by transport.
I have only minor comments as follows:
M & M section
P.3 l.116. Regarding to the histological analysis, which part of intestine were sampled? Please describe, because histological features are different according to the part of intestine.
P.4 l.170. Use of R needs citation below.
“R Core Team (2016). R: A language and environment for statistical computing. R Foundation for Statistical Computing, Vienna, Austria. URL https://www.R-project.org/.”
Results Section
Regarding to Figure 3C, is LZM liver lysosomal activity? Please put “(LZM)” after liver lysozyme in the legend for figure 3.
Author Response
Point 1: P.3 l.116. Regarding to the histological analysis, which part of intestine were sampled? Please describe, because histological features are different according to the part of intestine.
Response 1: We collected posterior section of the intestines and we have described the part of intestine we sampled for histological analysis at Line117.
Point 2: P.4 l.170. Use of R needs citation below.
“R Core Team (2016). R: A language and environment for statistical computing. R Foundation for Statistical Computing, Vienna, Austria. URL https://www.R-project.org/.”
Response 2: We have supplemented the reference at Line185. Thank you very much.
Point 3: Regarding to Figure 3C, is LZM liver lysosomal activity? Please put “(LZM)” after liver lysozyme in the legend for figure 3.
Response 3: We have added the LZM after lysosomal at Line247, thank you for the reminder.

Reviewer 2 Report
This MS measures several endpoints of fish health in largemouth bass juveniles exposed to simulated transport stress. The work is interesting as a first-pass look at identifying physiological changes in transported fish. I have 3 major concerns that lead me to suggest a major revision prior to publication:
-
My greatest concern is how the authors know the simulated transport conditions actually “stressed” the fish… there are few references in the M&M to back this up. Define “stress” and how you measure it.
-
There are a lot of inferences made between parameters the authors measure and an effect on fish health; these are rarely however backed up with data. For example, shortened intestinal villus length was observed in transport-stressed fish that was suggested would “greatly affect nutrient digestion and absorption” (line 339) without actually measuring nutrient metabolism. There are many other apparent causes and effects in the discussion (e.g., 4.3; 4.4: attempted links between gut microbiome species abundance and fish stress status) that are only suggested by the data. The authors should avoid making such definitive statements.
-
There needs to be more detail about experimental methods for several techniques used. For example, in line 39 and throughout, use accepted fish development terms like hatchling, fry and fingerling instead of seedling for juvenile fish. These terms will also specify juvenile age in the text for readers. Similarly, an age should be given for the fish (days after hatching) in section 2.1. They appear to be older than fingerlings, but this should be made clearer. I think the fish were lab-cultured, but give more info on the tank size, diurnal pattern, etc. Were the fish used a mix of males and females? If experimental fish were not sexed, could this change the results?
Other comments are below. If the suggested changes are made, the revised MS will be suitable for publication.
The introduction could have more of a general beginning to put the problem examined in context; e.g., Transportation… cultures. However, prolonged transportation periods stress fry in a number of ways…
2.2: was the bag size and water amount in each of them typical for transported fry (e.g., how many fingerlings are typically transported in each bag)? Need to explain why you chose this experimental setup. Include detail on bag size/volume, how temperature was maintained, light regime. How do you know that the conditions used in the experiment stressed the fish? The water quality endpoints are much below legislated values. For example, in British Columbia, Canada, nitrate levels below 3 mg/L are considered safe for aquatic life, and the maximum levels recorded in the experiment were 20-fold below this level. There is no method reference given in 2.2.
L102: 15 fish were randomly selected… does this mean 5 of the 15 fish were sampled from each of the 3 bags per treatment?
L136 and 144: what are the “correct” target bands on an agarose gel that allowed prokaryotic DNA and PCR products to pass the QC step?
L137: clarify what the universal primers used amplify… all prokaryotic DNA in the samples? Give a reference for the universal primers.
L146: how did you go from amplified DNA (PCR products) to an amplicon library? Was there a cloning step?
Fig 1C: DO is ~4x higher in “transported” fish versus controls. Why were DO levels so much higher in the simulated transport fish? This elevated DO could skew the results… for example, high DO could itself be stressful to the fish and cause the oxidative damage suggested in line 354.
L190: clarify “proportions”: did the vacuoles occupy 23% of the cross-sectional area of the hepatocytes?
Table 1: too many decimal places. Round values to nearest decimal place as appropriate. E.g., for the T0H column, top to bottom: 0.81+/-0.02, 39+/-5, and 391+/-7. Need the control values in the table as well and assume that controls were sampled at the same time. Including a “% of control” value for each of these makes comparison easier.
Fig 3: define all acronyms like LZM in legend.
Discussion 4.1: put the observed levels of N and DO in context… are they higher than recommended levels for aquarium-raised fish? At what concentration does the literature show fish suffer irreversible damage?
Author Response
Point 1: My greatest concern is how the authors know the simulated transport conditions actually “stressed” the fish… there are few references in the M&M to back this up. Define “stress” and how you measure it.
Response 1: Please excuse my unclear narration. Studies have shown that many reasons such as water quality changes, fish body bruising, starvation and so on may cause transport stress during transportation. We used automated shaker and the vibration frequency was set at 100 rpm simulate the actual transportation, and study have verified that the stable shake frequency is reasonable and can cause transport stress to fish. We have described it in more detail in the introduction Line40, and give reference in the M&M, Line105. We hope the modification can express the meaning more clearly, thank you for your concern.
Point 2: There are a lot of inferences made between parameters the authors measure and an effect on fish health; these are rarely however backed up with data. For example, shortened intestinal villus length was observed in transport-stressed fish that was suggested would “greatly affect nutrient digestion and absorption” (line 339) without actually measuring nutrient metabolism. There are many other apparent causes and effects in the discussion (e.g., 4.3; 4.4: attempted links between gut microbiome species abundance and fish stress status) that are only suggested by the data. The authors should avoid making such definitive statements.
Response 2: We have modification some definitive statements in the discussion (e.g., Line355; Line364; Line376; Line 379; Line412; Line465; Line474). Thank you for your reminder. In 4.4, we want to investigate the effects of transport stress and recovery on the gut microbiomes of juvenile largemouth bass. Previous studies have shown that gut microbes are closely related to host health. Therefore, we in line with fish stress status, analysis the reason of gut microbiome species abundance changes and assess the health status of the host rather than attempted links between gut microbiome species abundance and fish stress status.
Point 3: There needs to be more detail about experimental methods for several techniques used. For example, in line 39 and throughout, use accepted fish development terms like hatchling, fry and fingerling instead of seedling for juvenile fish. These terms will also specify juvenile age in the text for readers. Similarly, an age should be given for the fish (days after hatching) in section 2.1. They appear to be older than fingerlings, but this should be made clearer. I think the fish were lab-cultured, but give more info on the tank size, diurnal pattern, etc. Were the fish used a mix of males and females? If experimental fish were not sexed, could this change the results?
Response 3: The largemouth bass we used were 50 days of age with average body length 8.42 ± 0.44 cm, average weight 10.26 ± 0.32 g, and we have use fingerling instead of seedling for juvenile fish. The bass were cultured in a recirculating aquaculture system consist of 26 cylindrical circulation barrels with diameter 1.0 m, height 1.2 m, and the water used for aquaculture was filtered pond water. The breeding density were 1.3 g/L. We have reported in more detail in 2.1, Line88, Line90 and Line94. Thank you for your valuable advice. We used the fish with a mix of males and females for we try to simulate actual transportation in order to solve the problem of decreased vitality of largemouth bass after actual transportation, so, our experiment did not involve sex discrimination. On the other hand, we demonstrated that there is no grow difference in male and female growth before sexual maturity.
Point 4: The introduction could have more of a general beginning to put the problem examined in context; e.g., Transportation… cultures. However, prolonged transportation periods stress fry in a number of ways…
Response 4: Your opinion is very valuable, we have made a general beginning to complete the problem examined in our experiment in Line39.
Point 5: 2.2: was the bag size and water amount in each of them typical for transported fry (e.g., how many fingerlings are typically transported in each bag)? Need to explain why you chose this experimental setup. Include detail on bag size/volume, how temperature was maintained, light regime. How do you know that the conditions used in the experiment stressed the fish? The water quality endpoints are much below legislated values. For example, in British Columbia, Canada, nitrate levels below 3 mg/L are considered safe for aquatic life, and the maximum levels recorded in the experiment were 20-fold below this level. There is no method reference given in 2.2.
Response 5: The bag size is 40×80 cm, and the 1/3 volume is about 8 L of water, the transport density is close to 20 g/L, it was in agreement with the typically transport and this method has been widely used in fry or fingerling transportation. We set automated shaker at 100 rpm to simulate the actual transportation, and study have verified that the stable shake frequency is reasonable and can cause transport stress to fish. And we have given reference given in Line105. We set the air conditioning temperature to 22 degrees to ensure that the ring temperature is constant during simulated transportation, and avoid all light in the process. According to your proposal, we have supplemented these details in Line106. The water quality endpoints in our experiment are much below legislated values, but we think the transport stress comes from many factors, including water quality changes, fish body bruising, starvation and so on. Although the water quality endpoints are much below legislated values, they are more consistent with the characteristics of transport stress. This water quality index consistent with previous studies, and we have given reference in Line335 Thank you for your suggestion.
Point 6: L102: 15 fish were randomly selected… does this mean 5 of the 15 fish were sampled from each of the 3 bags per treatment?
Response 6: Yes, 5 of the 15 fish were sampled from each of the 3 bags per treatment, and we have made a supplementary according to your suggestion in Line107.
Point 7: L136 and 144: what are the “correct” target bands on an agarose gel that allowed prokaryotic DNA and PCR products to pass the QC step?
Response 7: Correctly sized target band about 700 bp were considered qualified samples. We have added it in Line 157.
Point 8: L137: clarify what the universal primers used amplify… all prokaryotic DNA in the samples? Give a reference for the universal primers.
Response 8: Yes, all prokaryotic DNA were amplified by the universal primers: v3-v4 region: 341F: 5’-CCTACGGGNGGCWGCAG-3’ and 805R: 5’-GACTACHVGGGTATCTAATCC-3’. We have given a reference for the universal primers in Line150.
Point 9: L146: how did you go from amplified DNA (PCR products) to an amplicon library? Was there a cloning step?
Response 9: We used Pacific Biosciences SMRTbellTM Template Prep kit 1.0 (Kapa Biosciences, Woburn, MA, USA) to prepare a sequencing library, we have made a supplementary in Line159. Thank you for the reminder.
Point 10: Fig 1C: DO is ~4x higher in “transported” fish versus controls. Why were DO levels so much higher in the simulated transport fish? This elevated DO could skew the results… for example, high DO could itself be stressful to the fish and cause the oxidative damage suggested in line 354.
Response 10: Transportation is carried out under high density, however, the air is difficult to meet the long distance transportation, which will cause the fish to die of hypoxia. Therefore, it is necessary to fill the bags with pure oxygen to meet the oxygen demand of transported fish. This makes us overlook the effects of hyperoxia on fish. We have reconsidered the hyperoxia possible impact on the possibility of generating oxidative stress in fish in 4.1, Line344. Thank you very much for your advice.
Point 11: L146: L190: clarify “proportions”: did the vacuoles occupy 23% of the cross-sectional area of the hepatocytes?
Response 11: Yes, “proportions” means the proportion of vacuoles in the cross-sectional area of the hepatocytes, and we have clarified it in 2.5, Line 139.
Point 12: Table 1: too many decimal places. Round values to nearest decimal place as appropriate. E.g., for the T0H column, top to bottom: 0.81+/-0.02, 39+/-5, and 391+/-7. Need the control values in the table as well and assume that controls were sampled at the same time. Including a “% of control” value for each of these makes comparison easier.
Response 12: We have rounded values to nearest decimal place as appropriate and calculation of control values in Table1, but we don’t add ‘% of control’ in the table for it may creates ambiguity (e.g. The liver vacuolar area in T0h is 0.81±0.02, and 0.37±0.04 in control group. It's a reasonable number but % of control about 200%, could not clearly expressed the experimental results). Thank you for your generous advice
Point 13: Fig 3: define all acronyms like LZM in legend.
Response 13: We have defined all acronyms in legend in Line 247, thank you for the reminder.
Point 14: Discussion 4.1: put the observed levels of N and DO in context… are they higher than recommended levels for aquarium-raised fish? At what concentration does the literature show fish suffer irreversible damage?
Response 14: We have put the the observed levels of water quality index in context. The ammonia-nitrogen and nitrite-nitrogen concentrations may lower legislated values, and there may be need a higher concentration for fish to suffer irreversible damage. But it significantly higher than culture level before experiment (ammonia-nitrogen concentrations about 0.01 mg/L and nitrite-nitrogen concentrations lower than 0.01 mg/L). Studies have shown that the concentration of ammonia nitrogen and nitrite in our experiment have affect on physiology and biochemistry of fish. At the same time, we think the transport stress comes from many factors, including water quality changes, fish body bruising, starvation and so on, many factors work together cause to transport stress. For another, this water quality index consistent with previous transport stress studies, and we have given reference in 4.1, Line335. Thank you for your suggestion.
Chai Y.H.; Peng R.B.; Jiang M.W.; Jiang X.M.; Han Q.X.; Han Z.R. Effects of ammonia nitrogen stress on the blood cell immunity and liver antioxidant function of Sepia pharaonis. Aquaculture 2021, 546, 737417.
Chand R.K.; Sahoo P.K. Effect of nitrite on the immune response of freshwater prawn Macrobrachium malcolmsonii and its susceptibility to Aeromonas hydrophila. Aquaculture 2006, 258, 150-156.

Reviewer 3 Report
The authors of the work undertook an interesting issue, the assessment of the impact of stress caused by transport on the physiological parameters of juvenile largemouth bass. The topic is important from both a scientific and practical point of view. The causes of stress caused by transport can be divided into mechanical and chemical. Chemical includes the impact of toxic metabolites formed in the tank, of which the authors focused on nitrogen compounds. Correctly described and presented results indicate that the fish showed signs of severe oxidative stress. Unfortunately, Authors completely missed high oxygen levels as a potential cause of oxidative stress and considered the increase in concentrations of nitrogen compounds as the only cause. In my opinion, in order to distinguish between these two causes (oxygen versus nitrogen compounds), additional, appropriately designed studies should be performed. If this is not possible, the discussion should be supplemented with a second thread (oxygen) and conclusions should take into account both possible causes. In addition, recommendations for future studies could be proposed.
Were the fish fed during recovery?
Were the daily changes in the bacterial population examined, i.e. immediately after feeding the fish in relation to 12-hour fasting? The 12-h group of fish was not fed for 24+12 hours. Were the fish in the control group also not fed for 36 hours?
What is the possible origin of Cyanbacteria in the guts? Has the change in water for transport influenced the changes in its occurrence?
Line 308. Effect of transport stress on dissolved oxygen… What do you mean? The title does not correspond to the content of the subchapter. The authors should discuss the possible impact of very high oxygen content in transport containers on the possibility of generating oxidative stress in fish. In the control samples, the oxygen level was almost 4 times lower.
Author Response
Point 1: Were the fish fed during recovery?
Response 1: Yes, the culture conditions described in 2.1 were applied to the control and recovery group, and we supplement it at Line96.
Point 2: Were the daily changes in the bacterial population examined, i.e. immediately after feeding the fish in relation to 12-hour fasting? The 12-h group of fish was not fed for 24+12 hours. Were the fish in the control group also not fed for 36 hours?
Response 2: We don’t exam the daily changes in the bacterial population. We did perform a 24-hour fasting treatment immediately after feeding. The 12-h group of fish did stop eating for 36 hours, but the fish in the control group were fed according to 2.1. We believe that there are many reasons for transport stress, such as water quality changes, fish body bruising and so on, feeding suspension may also be one of the reasons for transport stress. In this regard, we made a supplementary explanation in the introduction, Line39. So, we think that fasting is also one of the causes of transportation stress. In our study, we examined the bacterial population only in control group, 12-h group and recovery group, to investigate the effects of transport stress and recovery on the gut microbiomes of juvenile largemouth bass.
Point 3: What is the possible origin of Cyanbacteria in the guts? Has the change in water for transport influenced the changes in its occurrence?
Response 3: Thank you for pointing out the deficiencies about the Cyanbacteria. Cyanbacteria can pass into the gut combined with feed and may be passed out of the gut with the increase of fasting time. We don't think change in water for transport has influenced the changes in its occurrence. We think that Cyanophyta abundance in the 5-d recovery groups was markedly lower than in the control group may be arise from vitality decrease and eat less after transport stress. And we make a supplement at Line482.
Point 4: Line 308. Effect of transport stress on dissolved oxygen… What do you mean? The title does not correspond to the content of the subchapter. The authors should discuss the possible impact of very high oxygen content in transport containers on the possibility of generating oxidative stress in fish. In the control samples, the oxygen level was almost 4 times lower.
Response 4: I'm ashamed that I ignored the effects of hyperoxia. We have reconsidered the title and the hyperoxia condition possible impact on the possibility of generating oxidative stress in fish at Line344. Thank you very much for your advice.

Round 2
Reviewer 2 Report
Like I wrote in my first review, the authors should indicate that the N levels they measured in the stressed fish were still well (~20-fold) below those considered to be harmful to wild fish. In line 332, I suggest editing as follows:
In this experiment, while the ammonia-nitrogen and nitrite-nitrogen concentrations increased significantly (ammonia-nitrogen concentrations from 0.01±0.00 mg/L to 0.72±0.09 mg/L and nitrite-nitrogen concentrations from 0.01±0.00 mg/L to 0.13±0.02 mg/L) as the transport time increased, the detected N levels were still ~20-fold below those considered to be harmful to wild fish (with a suitable reference like https://www2.gov.bc.ca/assets/gov/environment/air-land-water/water/waterquality/water-quality-guidelines/approved-wqgs/bc_env_nitrate_waterqualityguideline_overview.pdf). Even so, elevated aquatic N levels may have increased the transport stress incurred by the largemouth bass, possibly owing to increases in fish metabolites.
Once this edit is included, this revised version is suitable for publication.
Author Response
Point 1: Like I wrote in my first review, the authors should indicate that the N levels they measured in the stressed fish were still well (~20-fold) below those considered to be harmful to wild fish.
Response 1: Thank you very much for your generous guidance for us. We have indicated that the N levels we detected were 20-fold below those considered harmful level. And we have enclosed your generous suggestion at line 332. Thank you again for your correction of our shortcomings.

Reviewer 3 Report
Thank you for your comments.
Author Response
This is what I should do. Thank you very much for your generous guidance for us.